# In Vitro Organotypic Systems to Model Tumor Microenvironment in Human Papillomavirus (HPV)-Related Cancers

**DOI:** 10.3390/cancers12051150

**Published:** 2020-05-03

**Authors:** Vincenza De Gregorio, Francesco Urciuolo, Paolo Antonio Netti, Giorgia Imparato

**Affiliations:** 1Interdisciplinary Research Centre on Biomaterials (CRIB), University of Naples Federico II, 80125 Naples, Italy; urciuolo@unina.it (F.U.); nettipa@unina.it (P.A.N.); 2Center for Advanced Biomaterials for HealthCare@CRIB, Istituto Italiano di Tecnologia, 80125 Naples, Italy; 3Department of Chemical, Materials and Industrial Production Engineering (DICMAPI) University of Naples Federico II, 80125 Naples, Italy

**Keywords:** human papillomaviruses (HPVs)-related cancers, tumor microenvironment (TME), 3D organotypic models, cervical cancers, anogenital cancers, oropharynx cancers

## Abstract

Despite the well-known role of chronic human papillomavirus (HPV) infections in causing tumors (i.e., all cervical cancers and other human malignancies from the mucosal squamous epithelia, including anogenital and oropharyngeal cavity), its persistence is not sufficient for cancer development. Other co-factors contribute to the carcinogenesis process. Recently, the critical role of the underlying stroma during the HPV life cycle and HPV-induced disease have been investigated. The tumor stroma is a key component of the tumor microenvironment (TME), which is a specialized entity. The TME is dynamic, interactive, and constantly changing—able to trigger, support, and drive tumor initiation, progression, and metastasis. In previous years, in vitro organotypic raft cultures and in vivo genetically engineered mouse models have provided researchers with important information on the interactions between HPVs and the epithelium. Further development for an in-depth understanding of the interaction between HPV-infected tissue and the surrounding microenvironment is strongly required. In this review, we critically describe the HPV-related cancers modeled in vitro from the simplified ‘raft culture’ to complex three-dimensional (3D) organotypic models, focusing on HPV-associated cervical cancer disease platforms. In addition, we review the latest knowledge in the field of in vitro culture systems of HPV-associated malignancies of other mucosal squamous epithelia (anogenital and oropharynx), as well as rare cutaneous non-melanoma associated cancer.

## 1. Introduction

Human papillomaviruses (HPVs) are double-stranded circular DNA epitheliotropic tumor viruses that are causally associated with all cervical cancers, as well as a significant fraction of several other human malignancies arising from the mucosal squamous epithelia of the anogenital tract (vaginal, vulvar, anal, penile), oral, and oropharyngeal cavity (mouth, throat, nasal sinuses, larynx, pharynx, salivary glands, and neck lymph nodes) and, infrequently, from the cutaneous epithelium (skin) [1,2,3]. According to 2018 data from the Global Cancer Observatory (GLOBOCAN), it is recognized that HPV contributes to more than 90% of cervical and anal cancers, approximately 78% of vaginal, and 25% of vulvar cancers, almost 53% of penile cancers, and 30% of head and neck cancers (HNCs), including oropharyngeal, oral cavity, and laryngeal cancers (30%, 2.1%, and 2.4%, respectively) (Figure 1) [4].

Recent findings indicate that about 60%–70% of the oropharyngeal cancers may be linked to HPV, although traditionally they were considered to be caused by tobacco and alcohol, or by a combination of these [5,6]. Based on their oncogenic potential, HPVs are classified as low-risk (LR) or high-risk (HR) viruses [7]. LR-HPVs can cause benign genital warts or laryngeal papilloma, whereas HR-HPV types are considered causally associated with nearly all cases of cervical cancer, other cancers of the lower female reproductive system and anus, as well as a high number of oropharyngeal cancers. The HR types include HPV16, 18, 31, 33, 35, 39, 45, 51, 52, 56, 58, and 59. Others are considered as potential HR including HPV 53, 66, 70, 73, and 82. The most virulent HR-HPV genotypes (HPV16 and HPV18) are major contributors to cervical cancer (50% of cervical squamous cell carcinoma are HPV16-positive and 35% of cervical adenocarcinomas are positive for HPV16 and HPV18), with 30% being caused by other HR-HPV types. Of note, most cervical intraepithelial neoplasia (CIN) lesions are HPV-related (HPV6/11/16/18 contribute to 23%–25% of CIN1, 38.4–39% to CIN2, and 58% to CIN3) [8]; HPV-negative CIN has also been reported. HPV16 is most commonly involved in the other HPV-induced cancers (e.g., oropharyngeal cancers ~25%) [9,10,11,12]. Recurrent respiratory papillomatosis is highly associated with LR-HPV6/11. HPVs are also responsible for a significant proportion of precancerous lesions of the vulva (vulvar intraepithelial neoplasia grades 2 and 3, VIN2/3), vagina (vaginal intraepithelial neoplasia grades 2 and 3, VaIN2/3), anus (anal intraepithelial neoplasia grades 2 and 3, AIN2/3), penis (penile high-grade squamous intraepithelial lesions), head and neck, as well as genital warts [13].

Although HPV infection is usually solved by the immune system and the vast majority of the virus infections are transient and asymptomatic, persistent HPV infections have an increased chance to induce epithelial cell abnormalities that can ultimately cause cancer [14,15]. HR-HPVs infect a wide range of epithelial sites, but cause cancer at these sites at different frequencies [16]. In detail, HPV initial infection can occur on monolayer squamocolumnar cells, or on squamocolumnar junction cells at the transformation zone regions of the cervix and anus, as well as on the reticulated epithelium of the palatine tonsil. However, HPV infection may also arise upon micro-abrasions of multi-layered epithelium [16,17]. HPV affinity to junctional tissues is due to the fact that the basal cells of the squamocolumnar transformation zone are particularly accessible and are thought to be more receptive toward HPV-mediated transformation [18]. On the other hand, HPV life cycle is closely linked to the differentiation program of the pluristratified epithelial host cell [19,20]. HPV may access dividing basal epithelial cells by falling down micro-abrasions in the epithelium and attaching to cells using common cell surface molecules [21]. The initial HR-HPV type infection determines low grade disease (low-grade squamous intraepithelial lesions, LSIL), due to inhibition of the normal differentiation in the lower third of the epithelium. The lesion may remain low-grade, regress, or progress to severe dysplasia or high-grade squamous intraepithelial lesions (HSIL). This latter stage may persist for several years or may progress from premalignant disease (CIN2/3) to invasive cancer, which, in some cases, leads to metastatic [22]. The relationship between HPV and the host genome may change during progression from premalignant to malignant phases of the disease [23]. The HPV genome integration in the host cell genome often occurs in HSIL, but episomal DNA is found in some cancers [24]. As a consequence, investigating the involvement of the HPV infection in the development of cancer is clinically and scientifically relevant. The progression of HPV replication and the early viral gene expression (E6 and E7) requires a highly-regulated differentiation program of stratified epithelia. In particular, HPV E6 and E7 oncoproteins partially inhibit and/or delay epithelial differentiation in the host cells via a variety of mechanisms, some of which involve the inactivation of pro-proliferative targets, such as retinoblastoma protein (pRb) and p53, promoting the epithelial cell proliferation and evading the apoptosis process [25,26]. Moreover, upon HPV infection, the stratified epithelium starts communication with the underlying stroma. HPVs interact predominantly with extracellular matrix (ECM) components during keratinocytes infection through the link with membrane-associated heparan sulfate proteoglycans, determining the HPV-infected epithelial cells invasion across the stromal barrier [27,28,29]. More recently, important and emerging roles during the HPV life cycle and HPV-induced disease of the matricellular proteins constituting the underlying stroma, or tumor microenvironment (TME), have become clearer [30]. For several years, the role of the TME in carcinogenesis has been underestimated, considering the stroma as a merely supportive scaffold upon which epithelial cells adhere, but it is now recognized that the stromal-to-HPV-infected-epithelial communication events play a key role in carcinogenesis [31]. Furthermore, stromal fibroblasts, a major cellular component of the connective tissues, also provide important signals in the development and progression of cancer, and it would be interesting to study their role in regulating the HPV life cycle and their presence in the neighborhood of HPV-induced lesions [32]. Of note, there is evidence that cancer-associated fibroblasts (CAFs), fibroblasts activated by paracrine mediators produced by cancer cells, may facilitate HPV-mediated carcinogenesis through a variety of mechanisms involving stromal-to-epithelial crosstalk [33]. Furthermore, the bidirectional communication between epithelial cells and the TME has been reported to affect tumor initiation and neoplastic progression to metastasis and drug resistance [34]. Researchers assume that, in response to this communication, the microenvironment interchanges contact through various stromal-to-epithelial signaling events involved in HPV-positive epithelial cell growth, disease initiation, and maintenance [35]. In vitro model systems able to replicate the interactions between HPV-infected tissue and the surrounding TME are required to an in-depth understanding of these phenomena (Figure 2). The aim of this review article is to summarize the most recent advancement in the field of tissue engineering regarding the development of the 3D organotypic model of HPV-associated disease, focusing firstly on cervical cancer disease, and then to other human mucosal malignancies-derived, spanning from the anogenital tract, oropharynx to cutaneous epithelium. This review also addresses the issue of the cross-talk between the stroma and its microenvironment on HPV-infected epithelia, emphasizing the need for most relevant human in vitro models to study the host-pathogen, as well as HPV-infected-TME interactions in cancer development.

## 2. HPV-Related Cervical Diseases

In developing countries, cervical cancer is still the second leading cause of cancer death, while in the western world the number of deaths continues to decrease thanks above all to the introduction of an early diagnostic examination [36]. One of the main risk factors for cervical cancer is HPV infection that is sexually transmitted. However, not all HPV infections cause cervical cancer. Most women who come into contact with such viruses are able to resolve the infection, with their immune systems, without subsequent consequences in their health. Finally, it has now been ascertained that only some of the over 100 types of HPV are dangerous from an oncological point of view, while the majority remain silent, or only give rise to small benign tumors called papillomas (also known as genital warts) [17]. Other factors can increase the risk of cervical cancer, including cigarette smoking, familiarity and genetic features, diet and obesity, as well as bacterial infections (e.g., Chlamydia trachomatis) [37,38,39]. Cervical cancers are classified according to the cells from which they originate in two types: squamous cell carcinoma, which accounts for 80% of cervical cancers, and adenocarcinoma, which accounts for about 15% percent of cervical cancers. Squamous cell carcinoma is a tumor that arises from the cells covering the surface of the exocervix while adenocarcinoma is a cancer that starts from the glandular cells of the endocervix. Adenosquamous carcinomas are known to be uncommon cervical tumors with a mixed origin [40]. The precursor lesions of cervical cancer are classified as a squamous intraepithelial lesion (SIL) or, an alternative term, cervical intraepithelial neoplasia (CIN), and are graded into three categories (I to III) depending on the extent of abnormal maturation: from low-grade lesions (CIN I) to dysplastic (CIN II) and severely dysplastic (CIN III). Viral genome integration has been proposed as an activation mechanism for progression from low- to high-grade lesions [41]. HPVs firstly infect basal epithelial cells, amplifying their genomes as low-copy-number, autonomously replicating in episomal form (establishment phase). Whereas, in the maintenance phase, the viral genomes are stably maintained at an almost constant copy number. The replication of the viral genomes occur during the S-phase, in synchrony with the host DNA replication. Moreover, in the productive or vegetative phase in vivo, which occurs only in highly differentiated cells, a high copy of numbers of the viral genomes are produced, and the packaging of viral capsids occurs only on the most highly differentiated layers of the epithelium [42,43,44].

### 2.1. In Vitro Model to Reproduce HPV Life Cycle in HPV-Related Cervical Cancers

The first models produced for investigating the HPV replication mechanisms were developed by transplanting infected foreskin explants into immune-compromised mice with the aim to propagate virion stocks from HR-HPVs (HPV16 and HPV18) [45,46]. However, although the animal model can provide considerable basic information on cervical lesion formation and regression, they have a number of limitations linked to the species specificity, different histological appearance, and epithelial tropisms of the HPVs [19,47]. In fact, HPVs are able to reproduce their entire life cycle only in human stratified epithelial cells and cause cancers at discrete epithelial sites for which straightforward in vivo models are missing [48]. The first attempts to produce the HPVs life cycle in vitro have met with little success. Researchers have hardly worked on the development of in vitro model systems that accurately mimic the mechanism of HPV infection in humans. Due to the correlation between the HPV life cycle and squamous epithelial differentiation, the most powerful in vitro models are represented by three-dimensional (3D)-organotypic epithelial tissues, known as “raft” culture [49,50]. Raft cultures have allowed normal keratinocytes to stratify and fully differentiate in air-liquid interface culture to produce a squamous full-thickness epithelial tissue when seeded on the top of a dermal equivalent, consisting of the fibroblast layer or fibroblast-populated collagen gel [51]. These 3D models have provided an environment permissive for recapitulating and modulating the infection program of cancer-associated HR-HPVs [52,53], which cause the majority of cervical cancers. Raft cultures were first developed by Asslineau and Pruniéras [54] and then improved by Kopan et al. [55], who have reproduced the entire HPV life cycle, including virus production, and have developed a dysplastic lesion similar to those observed upon in vivo HPV infection. In subsequent years, the HPV particles assembly has been studied with the use of in vitro-derived particles, such as virus-like particles (VLPs), pseudovirions (PsV), and quasivirions (QV) [56,57,58]. However, these techniques suffer from technical constraints for the production of “native” HPV virions in 3D organotypic culture. More recently, Conway et al. showed the “native” HPV virion production in a 3D organotypic model, obtained by immortalizing in vitro human foreskin keratinocytes (HFK) on 3T3 feeder layer, which allowed to deep investigate the HPVs’ assembly and maturation [59]. Infectious HPV progenies were isolated from such 3D organotypic models initiated with cells that maintain the HPV genome in the extrachromosomal replicative form [60]. In recent years, an increasing number of publications have reported the use of raft cultures produced with in vitro integrated human papillomavirus sequences in the cervical cancer cell line (Table 1). The latter are cervical carcinoma derived cell lines, such as SiHa cells, which contain an integrated HPV16 genome (HPV16, 1–2 copies/cell) [61], CaSki cells that contain an integrated HPV16 genome (600 copies/cell) as well as sequences related to HPV18 [62], and HeLa cells that contain HPV18 sequences cultured on collagen-populated-fibroblast gel [63,64]. C-33a, a pseudodiploid human cell line, was usually used as a control since these cell types are negative for human papillomavirus DNA and RNA [65]. Additionally, established squamous carcinoma cell line were seeded onto collagen plug to reproduce the dysplastic morphologies mimicking the pre-neoplastic lesions seen in vivo [52,66,67]. Other researchers reported the use of HPV-16 episome, containing normal immortalized human keratinocyte line (NIKS) that has been extensively cultured to study some aspects of HPV biology and transformation, particularly on raft culture [68]. As it is known, the physical state of the virus changes in the host cells from ‘episomal’ to ‘integrated’ in the polyclonal premalignant lesion, potentially promoting the disruption of human gene loci that are relevant to cancer pathogenesis [69]. In this perspective, Stanley et al. produced a non-tumorigenic human cervical keratinocyte cell line, i.e., W12, from a polyclonal culture of cervical squamous epithelial keratinocytes naturally infected with HPV16, which were derived by explant culture of a low grade-SIL, histologically diagnosed as CIN I [70]. They demonstrated that, at early passages, these ‘parental’ W12 cells are able to maintain a stable genome and phenotype. In 3D organotypic culture, W12 cells maintain the HPV16 genome as episomes at about 100 copies per cell, recapitulating a low grade-SIL. In long-term culture, W12 cells lose these properties and phenotypically progress to high grade-SIL and then to squamous cervical carcinoma (SCC) with HPV16 DNA integration, recapitulating the host events associated with cervical carcinogenesis in vivo [71]. On the other hand, primary HFK isolated from clinical circumcision at a low passage, that stably maintaining episomal HPV genome, have been frequently used in the laboratory due to the ease of isolation and the high cell yield [72,73,74,75,76,77]. Notwithstanding, an important aspect to take into account when the researcher decides to build a 3D organotypic model is the use of cells derived from the organs that they intend to study. The epithelia of different organs have a dissimilar differentiation program with or without epithelial keratinization and with different cytokeratins expression at the epithelial layers [78,79]. Moreover, HR-HPVs differently re-program the keratinocytes to express or delay the epithelial differentiation by altering the expression pattern of specific differentiation markers [80]. These phenomena appear to be unique in every anatomical site. For this reason, it is, however, incorrect to use HFK to reproduce cervix uterine models in vitro, although they are more available than cervical tissues. Such models do not reproduce the histological features of the cervical tissue but display the distinctive cutaneous skin morphology. In addition, HPV-infected epithelial mucosa of cervix uterine showed differences from skin particularly in terms of hormone responsiveness and immunological activators [81]. Preferably, 3D organotypic models have to be developed from primary cervical keratinocytes derived from healthy biopsies, and infected by introducing, experimentally, the HPVs or by propagating cervical keratinocytes isolated from naturally HPV-infected biopsies from cervical lesions [17,82]. In this perspective, some scientific reports have established a patient-derived cell culture system by using fresh cervical biopsies as a more accurate alternative to traditional cervical cancer-derived cell lines [83,84]. In detail, De Gregorio et al. developed an organotypic cervix model by using primary cervical cells, obtaining a complete ectocervical epithelium that showed all the characteristic epithelial differentiation markers when seeded onto a complex and auto-produced ECM [83]. Recently, Villa et al. devised a protocol to isolate and resuscitate (after freezing) cervical keratinocytes to model organoid culture [84]. These models closely resemble the in vivo stratified epithelium and may be useful for investigating the complex molecular mechanism of cervical neoplastic transformation related to persistent HPV infection.

### 2.2. In Vitro Organotypic Systems to Model Tumor Microenvironment in HPV-Related Cervical Cancer

It is now established that the stromal microenvironment contributes to tumorigenesis in HPV-related cervical cancer and that the TME sends signals that guide growth, tumor progression, and the formation of metastases, as well as resistance to anticancer drugs [85,86]. CAFs are known to have an active role in tumorigenesis providing relevant signals in the development and progression of the pathology through the release of growth factors that guide ECM remodeling and angiogenesis [87,88]. In addition, alterations in the production of ECM proteins, as well as ECM remodeling enzymes, can influence the stroma by modulating the carcinogenic potential of adjacent epithelial cells [87]. Emerging evidences suggest that a bidirectional crosstalk between HPV-positive epithelia and the underlying stroma occur during cancer progression [88,89]. In recent years, cervix models have been developed by culturing epithelial cells onto human foreskin or mouse 3T3-J2 fibroblasts (Table 1) to generate feeder layers as stromal equivalent, emphasizing the crucial role of the fibroblasts in the epithelial cell culture and propagation [90]. Some researchers have also highlighted the fibroblasts feeder layers involvement in promoting homeostasis and proliferation of the cervical epithelial cells by direct contact-dependent and/or indirect paracrine signaling, including soluble factors, such as growth factors and cytokines, as well as ECM components [90,91]. In this direction, researchers demonstrated that fibroblasts enhance, specifically, the HPV16- and HPV18-positive cervical epithelial cells growth and inhibit the normal epithelial cells growth by a ‘double-paracrine’ epithelial-stromal signaling mechanism involving, e.g., interleukin-1 proteins production [92,93]. In addition, fibroblasts feeder layers are also required for the HPV genome maintenance as extrachromosomal episomal form in HPV-infected cells, such as W12 [94]. More complex 3D cervix-like constructs, consisting of foreskin fibroblasts embedded within some matrices, such as rat tail fibrillar collagen, polymeric scaffolds, or de-epidermalized matrix, were developed in order to promote the epithelial cell stratification and provide support comparable to the extracellular matrix [49]. Recent studies pointed out that ‘human’ stromal fibroblasts promoted much more epithelial invasion than ‘mouse’ fibroblasts in HPV-positive epithelia grown on raft culture, demonstrating the need for associating tissue-specific cells with specific tissue to better reproduce the native microenvironment for in vitro tumor modeling [95,96]. Moreover, 3D organotypic cultures built up from cervical tissue derived from healthy or tumor/cancer-associated keratinocytes and fibroblasts, may be used to correctly resemble, in vitro, the native morphologic and histologic cervical features, and also to better elucidate the interactions between stromal and epithelial compartments in the carcinogenetic process [97,98]. In a pioneering work, a 3D organotypic cervical model was developed by using tissue-specific cells from different organs, among this uterine cervix, to replicate more faithfully the native tissue, showing the epithelial invasion through the basement membrane during tumor progression [99]. Other studies highlighted the crucial role of the stromal-derived factors in promoting epithelial invasion in cervical cancer [100]. Furthermore, another important aspect to take into account when fabricating the engineered tissue equivalents is organ complexity. Reconstructed models should provide the same structure and composition as the cervical mucosa, including, for example, the presence of both collagen and non-collagenous proteins (glycosaminoglycans, etc.) that, in vivo, are involved in the ECM remodeling occurring during epithelial mesenchymal transition (EMT) process [101,102]. Although the ECM represents a non-cellular component within tissues, to which, generally, a supporting role is ascribed, it has been recently recognized that ECM has a fundamental functional repository role for several factors that dynamically modulate the TME [103]. Some molecules are deposited into the ECM and remain latent until activated. Among these, the Matrix Metalloproteinases (MMPs) hollow out space in the matrix allowing cells to migrate by degrading ECM components [104]. MMPs also regulate the epithelial cell migration and interactions within the stroma [105]. To demonstrate this, Fullar et al. investigated the action of the HPV16 on the MMPs produced by epithelial cells and fibroblasts during EMT and carcinogenesis processes [106]. Other researchers highlighted the induction of the ECM remodeling in HR-HPV positive models [107]. It is well known that, in vivo, the cervical stroma undergoes a controlled remodeling by quantitative/qualitative protein changes mediated by specific enzymes and the dysregulation of the ECM composition, structure, stiffness, and abundance affects the pathophysiological tissue status, contributing to several pathological conditions, such as invasive cancer [103,108]. Recently, cervical microtissues have been used as functional units for the manufacture of an endogenous cervical stroma, with stromal characteristics comparable to those of the native cervix. This complex 3D cervix tissue equivalent was provided by a specialized ECM microenvironment featured by an autologous and responsive ECM and an auto-produced basement membrane. Indeed, the 3D completely scaffold-free ECM was able to guide the formation of fully differentiated and stratified epithelium, establishing the correct cross-talk between stroma and epithelium [83]. Organotypic tissues that recapitulate in vitro the composition and structural organization (epithelial stratification, functional basement membrane, fibroblast populated stroma, complex ECM) of their native counterpart represent a new model for studying tumor progression and evaluating combined therapies in non-animal models (Table 2). Consistent with this finding, unpublished work from our laboratory indicates the importance of stromal-to-epithelial communication in guiding the mesenchymalization in HPV-positive epithelia. The crucial role of the cervical cancer-associated fibroblasts, as well as the stromal microenvironment in the biochemical changes that enable the epithelial cells to assume a mesenchymal cell phenotype, were evidenced by analyzing EMT markers expression. Altered expression of the adhesion molecules, the collagenous and non-collagenous proteins as well as the ECM remodeling have been further highlighted. Finally, an up-regulation of the gene expression of late viral proteins on HPV-positive epithelial cells cultured on a diseased cervical model was also found. The epithelium-to-stroma and stroma-to-epithelium crosstalk may be the mechanism through which the viruses manipulate their environment and vice-versa, in the case of tumor-associated viruses, contributing to carcinogenesis [27]. It’s also noted that the carcinogenesis of cervical carcinoma implies an intricate interplay of neoplastic, HPV-infected epithelial cells, and stromal tissue including non-tumoral cell types [109]. The HPV-positive epithelium and stromal cells (CAFs, endothelial cells, immune cells and neuronal cells) communicate with HPV-infected epithelium through the exchange of growth factors (Transforming growth factor beta (TGF-β), Vascular endothelial growth factor (VEGF), Heparin-binding EGF-like growth factor (HB-EGF), Epidermal growth factor (EGF)), cytokines (Interleukins (ILs), chemokine (C-X-C motif) ligand 1 (CXCLs) and CC chemokines (CCLs)), neurotransmitters, ECM molecules (MMPs) and other molecules (macrophage colony-stimulating factor (M-CSF), Granulocyte-colony stimulating factor (G-CSF)), leading stromal remodeling, cancer cells proliferation and angiogenesis processes [27]. A thorough mapping of the non-tumoral cell types that populate the TME is critical to understand their unique roles in tumor biology. Interestingly, tumor innervation is associated with worse clinical outcomes in several solid cancers [110,111], emphasizing nerves as microenvironmental factors that may contribute to tumor progression. Scientific reports suggest that carcinogenesis alters cervical innervation, demonstrating the role of the HPV-positive cervical cancer cell lines in effectively stimulate neurite outgrowth [112]. Furthermore, an essential component of the tumor-associated stroma is the vasculature, composed of blood and lymph vessels [109]. The induction of angiogenesis is an early event in cervical carcinogenesis [113]. In details, in low-grade lesions, there is an increase in the number of capillaries in the cervical stroma underlying the dysplastic epithelium. In high-grade lesions there is an additional increase in the number of vessels that appear to be organized into a dense micro-vascular array in close relation to the overlying neoplastic epithelium. Furthermore, in some high-grade lesions, stromal vascular papillae are formed that reach towards the surface of the epithelium [114]. Over the past years, the endothelial cells have been used in vitro as feeder for keratinocytes to support epithelial cell differentiation [32]. Other studies reported the HPV-dependent angiogenic response in terms of proliferation and migration of the endothelial cells when cultured with conditioned media from HPV positive cells [115]. In this perspective, a further implementation of the complexity of the 3D organotypic cervix models may be to insert non-tumor cells (endothelial vascular cells and/or neuronal cells) at the stromal level to study the interaction between stroma and their adjacent complex TME [116,117]. Furthermore, immune cells infiltrate, such as macrophages, and T-cells as potential anti-cancer cellular weaponry may also greatly implement the 3D models’ complexity [118,119] (Figure 3). Finally, the reproduction of the microbial species of the cervical microbiota may be a step forward for the reproduction of models that faithfully mimic the native tissue [120,121].

## 3. HPV-Related Human Malignancies Arising from Mucosal Squamous Epithelia: Head and Neck and Anogenital-Tract Cancers

### 3.1. Head and Neck Cancers

Head and Neck cancers (HNCs) represent a heterogeneous group of tumors that include cancer of squamous epithelial cells of the oral cavity (mouth and throat), nasal cavity (sinuses), larynx and pharynx, salivary glands and lymph nodes in the neck [3]. Connective tissue cells such as CAFs promote squamous cell carcinoma proliferation, invasion, and metastasis, on the contrary lymphocytes are rarely involved in HNCs (about 10%) [124,125,126]. A large number of HPV-associated HNCs are caused by HPV16 and, although are hard to treat, they have favorable prognosis compared to HPV-negative HNCs. Moreover, 3D organotypic models provide a useful tool to evaluate the sensitivity of HPV-positive and HPV-negative cancer cells exposed to different therapeutic strategies to identify potential druggable targets for tailored therapy [127]. Among the heterogeneous group of Head-Neck cancers, HPV-related oropharyngeal cancers, that have a favorable prognosis compared to HPV-unrelated tumors, were deeply studied to have a better understanding of the HPV role in exacerbating malignant phenotypes [126]. In this direction, organotypic raft cultures, which included immortalized oral/oropharyngeal squamous carcinoma cell lines seeded on submucosal equivalents consisting of type I collagen and normal human oral fibroblasts, was used to study key characteristics of cancer [128,129]. Other researchers have developed an oral cancer equivalents system prepared with decellularized human dermal tissue that allowed to study the epithelial stratification and invasion beyond the basement membrane into underlying connective tissue [130]. In another study, Dalley et al. reported the cancer stem cells involvement in the progression of oral dysplasia to squamous cell carcinoma by developing 2D monolayer or 3D organotypic culture of normal, dysplastic, and squamous cell carcinoma-derived oral cell lines. The researchers demonstrated the usefulness of the 3D human oral mucosa equivalent on the detection of the hierarchical expression of cancer stem cell-associated markers (CD44, p75NTR, CD24, and ALDH), information that cannot be revealed in 2D monolayer [131].

Tonsillar carcinomas are among the most frequent HNCs, arising in the reticulated epithelial cells of the crypts with immune cell infiltrations. HR-HPV16 persistent infections are related to oropharyngeal carcinogenesis [80]. However, compared to cervical cancer, little is known about how HPV drives the pathogenesis of oropharyngeal cancer. HPV could establish a productive or abortive infection in keratinocytes of the tonsil crypt, or progresses to cancer through a neoplastic phase, as occurs in cervical HPV infection [132]. In oropharyngeal cancer, the HPV DNA is more frequently found un-integrated and may include novel HPV-human DNA hybrids episomes [127]. Meyers’ group recently reported the usefulness of the organotypic raft culture (composed of immortalized primary human tonsil and HFK cell lines persistently infected with HPV16 seeded onto collagen matrices consisting of rat-tail type 1 collagen and containing J23T3 feeder cells) to investigate the life cycle of HPV16 in oral (tonsil) epithelial tissue vs. genital (foreskin) tissue focusing on the titers, infectivity, and maturation of HPV16. They demonstrated that, although some aspects of the HPV16 replication are overlapped in foreskin and tonsil tissue, there are significant differences related to the maturation and final structure of the virions when grown in the two different tissues [133]. In addition, other works reported more complex 3D co-culture systems of tonsil keratinocytes, with immune components providing a more realistic in vitro model [134], since the tonsil has significant infiltration of lymphoid cells into the crypt that potentially explain its improved prognosis [135].

### 3.2. Anogenital HPV-Associated Cancers in Males and Females

To reproduce in vitro anogenital HPV-associated cancers in males and females, the main models developed are the 3D organotypic culture that allows to accurately mimic the life cycle of several cancer-associated HR-HPV types. In the previous sections, we focused on the culture system used to study HPV-related cervical cancer; here, we revised the in vitro model mimicking other organs of the human anogenital apparatus, both in male and female.

#### 3.2.1. Vulva and Vagina

Lower genital cancers are frequent malignancies in women prevalently associated with HR-HPVs infection. To date, some studies have reported that vulvar squamous cell carcinoma (VSCC) have different pathological pathways related, or not, to HPV infections. HPV-related VSCC with profound cellular atypia and basaloid/warty histology and VSCC non-dependent to HPV infection with basal atypia and keratinized histology [136,137,138]. In addition, VSCCs are also associated with chronic inflammatory dermatitis, such as vulvar lichen sclerosus [139]. Limited in vitro models have been developed for better understanding the biology and development of VSCCs. Some researchers have isolated and characterized primary cells from VSCC and normal vulvar tissue adjacent to the tumor in order to develop 3D organotypic and/or in vivo xenografts models [140,141,142]. Vaginal cancers are uncommon diseases that are also related to HPV infection. Most organotypic models have been developed for the ex vivo study of HIV infection [143]. Few works have reported the development of raft cultures obtained by seeding vaginal cells able to study microRNA biomarkers of oncogenic HPVs infections [144].

#### 3.2.2. Anus

Anal cancer is closely related to high-risk HPV persistent infection that occurs in the anal transformation zone, similar to what happens in the uterine cervix [145]. HPV infection leads to the development of an HSIL lesion that can then evolve to invasive carcinoma. Only a few models have been developed due to the lack of immortalized HPV-positive anal epithelial cell lines [10,146]. Among this, Wechsler et al. reported a novel in vitro model of anal cancer pathogenesis using the first HPV-16-transformed anal epithelial cell line (AKC2 cells) that have a poorly differentiated and invasive phenotype in three-dimensional raft culture. In this 3D contest, AKC2 cells express all three HPV-16 oncogenes (E5, E6, and E7) involved in anal cancer progression [147].

#### 3.2.3. Penis and Penile Urethra

For several years, the foreskin cell line derived from circumcision has been used to study various HPV-related cancer due to the high availability of this skin fold. The penile urethra, that consists of pluri-stratified squamous cells without keratinization process alike the foreskin or the glans, is also routinely targeted by sexually transmitted viral pathogens such as HIV infection in the mouse model [148]. Limited studies reported the organotypic raft culture of prenatal genital tubercle to investigate the direct effects of the hyperestrogenic state on fetal mouse penile and urethral development [149].

## 4. Other HPV-Related Cancers (Non-Melanoma Skin Cancer)

Among cutaneous malignancies, the non-melanoma skin cancer (NMSC) involves fair-skinned populations and are mainly correlated to solar ultraviolet irradiation [150]. However, this malignancy appears to be also related to HPV infection [151]. Current findings report studies on the critical role of cutaneous HPV infection as a co-factor in association with genomic and mitochondrial mutations induced by ultraviolet irradiation in NMSC development in simplified in vitro models [152]. Other researchers have focused on the study of the HPV transformation mechanisms, as well as the epithelial invasion in the NMSC, by using an in vitro skin-equivalent organotypic model [153]. These models better reproduce the terminal differentiation of the epithelial cells and also the migration and invasion through the underlying dermis after HPV infections.

## 5. Conclusions

In this review, we outlined the current knowledge on the HPV-related cancers modeled in vitro from the simplified ‘raft culture’ to complex 3D organotypic models focusing on HPV-associated cervical cancer disease platforms and we also reviewed the in vitro culture systems of human HPV-associated mucosal malignancies from the anogenital tract, oropharynx, and cutaneous epithelium. We highlighted the importance of using multicellular models, which involve the use of compartment-specific cell types at both the epithelial and stromal level, and a complex ECM capable of remodeling to fully reproduce the histomorphological features of the tissue in vivo. This review also addresses the issue of the cross-talk between the stroma and its microenvironment on HPV-infected epithelial, emphasizing the need for most promising in vitro models to study host-pathogen, as well as HPV-infected-TME interactions in cancer development in humans.

## Figures and Tables

**Figure 1 cancers-12-01150-f001:**
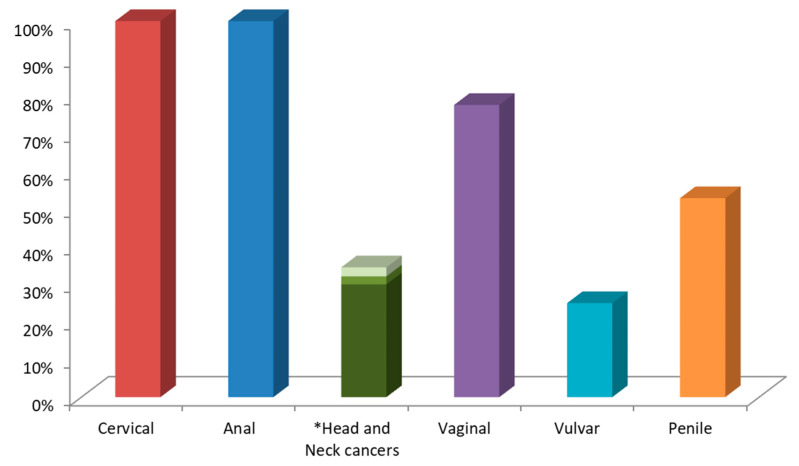
Representation of the estimated number percentage of human papillomavirus (HPV)-associated cancers vs. different cancer subsites. *Head and Neck cancers include oropharyngeal, oral cavity and laryngeal cancers.

**Figure 2 cancers-12-01150-f002:**
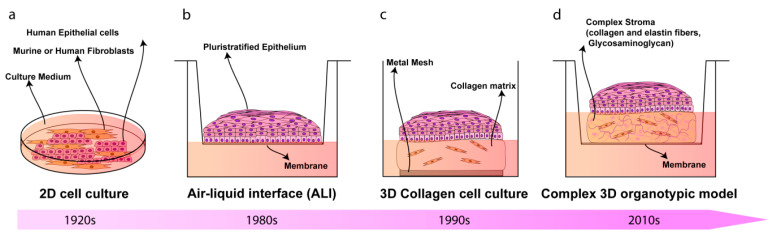
Timeline representing the evolution of in vitro culture systems for the study of HPV-related diseases. (**a**) Two-dimensional (2D) cell mono- or co-culture models have been used since the early 1900s; (**b**) in the 1980s, air-liquid interface (ALI) cultures were developed; these models consist of the epithelial cells seeding onto a semi-permeable membrane, allowing the formation of the epithelial strata; (**c**) three-dimensional (3D) collagen cell culture represented the first 3D model that reconstructs both epithelium and stroma and were developed in 1990s: primary or immortalized cells are cultured for 2–3 weeks onto the fibroblasts-feeder-layer or on collagen matrix populated with fibroblasts, intending to mimic the lamina propria. The epithelial cells, then, are induced to stratify and differentiate; (**d**) complex 3D organotypic models built up from tissue explant-repopulated-matrix, de-epidermized culture or endogenously produced extracellular matrix (ECM) was developed in the 2010s. These models provide a complex ECM with a well-differentiated epithelium that physiologically resembles the native tissue.

**Figure 3 cancers-12-01150-f003:**
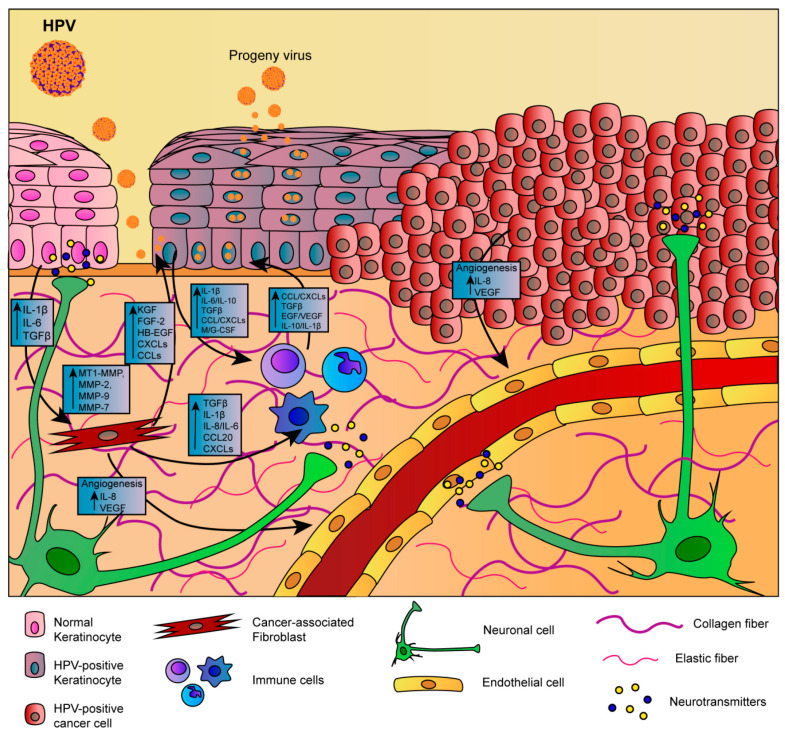
Diagrammatic representation of the bidirectional cross-talk between HPV-positive epithelium and tumor microenvironment (TME). HPV-positive epithelium and stromal cells (CAF, endothelial cells, immune cells and neuronal cells) communicate with HPV-infected epithelium through the exchange of growth factors (TGF-β, VEGF, HB-EGF, EGF), cytokines (IL, CXCLs and CCLs), Neurotransmitters and ECM molecules (MMPs) and other molecules (M-CSF, G-CSF). Large black arrows represent the bidirectional communication between HPV-positive epithelium and stromal cells. Small black arrows in the squares represent an increase (up arrow) of specific factor. CAF = Cancer-Associated Fibroblast; IL = Interleukins; CXCs/CCLs = Chemokines; TGF-β = Transforming Growth Factor-beta; EGF = Epidermal Growth Factor; HB-EGF= Heparin-binding EGF-like growth factor; FGF-2 = Fibroblast Growth Factor-2; VEGF = Vascular Endothelial Growth Factor; M-CSF = Macrophage Colony-Stimulating Factor; G-CSF= Granulocyte-Colony Stimulating Factor; MMPs = Metalloproteinases.

**Table 1 cancers-12-01150-t001:** Summary of different cell types used to develop tissue engineered cervical mucosa.

Cell Lines	HPV Types	Physical State	References
Epithelial Cells			
SiHa	HPV16	Int.	[61,64]
CaSki	HPV16/18	Int.	[62,64]
HeLa	HPV18	Int.	[63,64]
W12	HPV16	Int./Epi.	[70,71]
C-33a	-	-	[65]
NIKS	HPV16	Epi.	[68]
Primary HFK	HPVs	Int. or Epi.	[72,73,74,75,76,77]
Primary HCK	HPVs	Int. or Epi.	[82,83,84]
Stromal cells			
3t3J2 fibroblasts	-	-	[90]
Primary HFF	-	-	[91,92,93]
Primary HCF	-	-	[83,84,95,106]

Int. = integrated HPV genome; Epi. = episomal HPV genome; HFK = human foreskin keratinocytes; HCK = human cervical keratinocytes; HFF = human foreskin fibroblasts; HCF = human cervical fibroblasts; NIKS = normal immortalized human keratinocyte line.

**Table 2 cancers-12-01150-t002:** Summary of organotypic cervical models pointing out their advantages and limitations.

Cell Culture Systems	Advantages	Limitations	References
2D cell culture 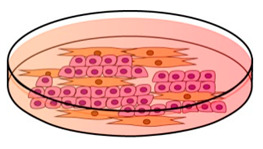	Simplified model/ Possibility of making co-culture/ Cost effective/ Easy to use	Inability to reproduce HPV life cycle/ Short time culture/ Low biological relevance/	[56,57,58,61,62,63,65,66,70,106]
Air-liquid interface culture 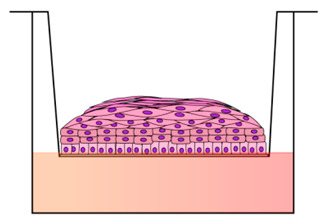	Recapitulate pluristratified epithelium/ Cell-to-cell-interactions/ HPV genetic studies/ Convenient/ Easy to use	Lack of connective tissue/ Genetic manipulation epithelial cell-dependent/	[122,123]
3D collagen cell culture 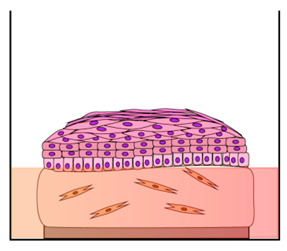	Resemble the epithelial architecture and differentiation/ Cell-to-ECM signaling/ More complex culture system/	Hydrogel composition differs to real ECM/ HPV genetic studies epithelial cells-dependent/ Added expensive/	[49,50,51,52,54,55,59,60,64,67,68,71,74,75,76,77,82]
Complex 3D organotypic models 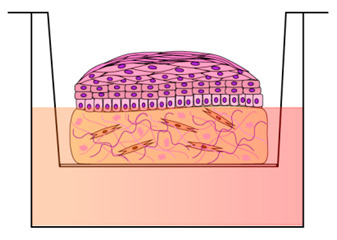	More accurate culture systems/ Cell-virus-ECM interactions/ ECM complexity/ Patient-specific models/ Long-time culture (3–4 weeks)/ Useful drug testing platform/ Useful for Invasion studies/	Differences between specimens/ Endpoint assays (genomic, proteomic, metabolomics) dependent of cell culture used/ More expensive/	[83,84,95,99]

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
