# Peer review of "In Vitro Organotypic Systems to Model Tumor Microenvironment in Human Papillomavirus (HPV)-Related Cancers"

_cancers, 2020, doi:10.3390/cancers12051150_

Round 1

Reviewer 1 Report

This an interesting review on the use of in vitro models to study HPV infection and replication as well as HPV-induced carcinogenesis. It covers the earliest development starting in the 80s till most recently developed systems. Besides studies related to cervical cancer also studies performed on other HPV-induced cancers are discussed.

Please provide a reference for 1 and align the percentages with the latest WHO/IARC data.

Lines 54-55: this sentence is not correct: persistent HPV does not induce skin injury, it may induce dysplasia. Please check with reference 10 on how to describe HPV-induced carcinogenesis.

Same for lines 62-66: please make sure this is correct.

Line 127: what is meant with maintenance phase? Does this concern a productive infection. Please make sure that the description in lines 127-130 is correct.

It will be helpful to make a clear distinction between initial HPV infection (often upon micro-abrasians of multi-layered epithelium, but may also be monolayer squamocolumnar cells), virus production and viral transformation in the intorduction. These are different processes.

Such a distinction is made when discussing the various models. However, little information is provided on the use of these models to study the carcinogenic  process in epithelial cells. It is mainly restricted to the interaction with the TME. So please extend a bit more to the carcinogenic process. 

Lines 160 and below: The term HPV-immortalized cells is confusing when discussing cancer cells. In general HPV-immortalized cells are referred to as in vitro immortalized keratinocytes.

Line 180: HFKs are usually isolated from circumcision rather than tissue biopsies. Please make sure the description is correct.

The listing of the cervical cancer models is very well appreciated. However I would like to have them summarized in a Table as well.

The review requires some language editing

Author Response

Reviewer 1

This an interesting review on the use of in vitro models to study HPV infection and replication as well as HPV-induced carcinogenesis. It covers the earliest development starting in the 80s till most recently developed systems. Besides studies related to cervical cancer also studies performed on other HPV-induced cancers are discussed.

Thank you the Reviewer for appreciating the quality of the review.

Please provide a reference for 1 and align the percentages with the latest WHO/IARC data.

We thank referee for his/her observation, and we replace Figure 1 (Line 45 in the revised version) with latest WHO/IARC dara including the estimated numbers of infection-attributable cancer cases in 2018 in order address the referee concerns.

Lines 38-42 in the revised version: “According to Globocan 2018 data, it is recognized the HPV contribution to prompt more than 90% of cervical and anal cancers, approximately 78% of vaginal and 25% of vulvar cancers, almost 53% of penile cancers, and 30% of Head and Neck cancers (HNCs) including oropharyngeal, oral cavity and laryngeal cancers (30%, 2.1% and 2.4%, respectively)”.

Line 47-49 in the revised version: Figure 1. Representation of the estimated numbers percentage of HPV-associated cancers vs different cancers subsites. *Head and Neck cancers include oropharyngeal, oral cavity and laryngeal cancers.

Lines 54-55: this sentence is not correct: persistent HPV does not induce skin injury, it may induce dysplasia. Please check with reference 10 on how to describe HPV-induced carcinogenesis.

We thank referee for his/her observation, and we adjusted the sentence in the revised version of the revised review.

Lines 84-87 in the revised version: “Although HPV infection is usually solved by the immune system and the vast majority of the virus infections are transient and asymptomatic, the HPV persistent infection have an increased chance of acquiring epithelial cell abnormalities that ultimately can cause cancer [14, 15]”

Same for lines 62-66: please make sure this is correct.

We thank referee for his/her observation and we reworked the sentence in the revised version of the review.

Lines 120-125 in the revised version: “Moreover, upon HPV infection, the stratified epithelium starts communication with the underlying stroma. HPVs interact predominantly with extracellular matrix (ECM) components during keratinocytes infection through the link with membrane-associated heparan sulfate proteoglycans, determining the HPV-infected epithelial cells invasion across the stromal barrier [27-29].  

Line 127: what is meant with maintenance phase? Does this concern a productive infection. Please make sure that the description in lines 127-130 is correct.

We thank referee for his/her observation and we accurately adjusted the sentence in the revised version of the review.

Lines 188-192 in the revised version: Whereas, in the maintenance phase, the viral genomes are stably maintained at an almost constant copy number. The replication of the viral genomes occurs during S-phase in synchrony with the host DNA replication.”  

It will be helpful to make a clear distinction between initial HPV infection (often upon micro-abrasians of multi-layered epithelium, but may also be monolayer squamocolumnar cells), virus production and viral transformation in the intorduction. These are different processes.

We thank referee for his/her constructive observation and we completely re-manage the Introduction section in order to address the Referee concerns.

Lines 84-100 in the revised version: “Although HPV infection is usually solved …….. endocytosis [21].

Such a distinction is made when discussing the various models. However, little information is provided on the use of these models to study the carcinogenic  process in epithelial cells. It is mainly restricted to the interaction with the TME. So please extend a bit more to the carcinogenic process.

We thank the referee for his/her observation and we have enriched the Introduction section in the revised version of the review including the details of the carcinogenic process in epithelial cells.

Lines 100-112 in the revised version: “The initial HR-HPV types infection determines ……..are selected [24]”.

Lines 185-194 in the revised version: “During the productive infection in vivo, …….the epithelium [42-44] .“

Lines 160 and below: The term HPV-immortalized cells is confusing when discussing cancer cells. In general HPV-immortalized cells are referred to as in vitro immortalized keratinocytes.

We thank the referee for his/her observation and we correct the main test of the revised version of the review. Lines 222-225, 227-229, 234-237 in the revised version

Line 180: HFKs are usually isolated from circumcision rather than tissue biopsies. Please make sure the description is correct.

We thank the referee for his/her observation and we corrected the sentence in the revised version of the review. Lines 251-253 in the revised version

The listing of cervical cancer models is very well appreciated. However, I would like to have them summarized in a Table as well.

In order to address the Reviewer concerns, we added a new table (Table 3, Line 401 in the revised version) in which are reported the list of cervical cancer models and we also added their advantages and limitation.

The review requires some language editing

We revised the new version of the review in order to improve the readability of the manuscript and we also improve the English quality.

Reviewer 2 Report

I enjoyed reading this review of in vitro methods for modelling the tumour microenvironment in HPV-associated cancers and I think it will be a very useful introduction for those who have maybe worked largely with 2D culture but are looking for more physiologically relevant systems. Overall, I found it very informative and clearly the authors have made substantial contributions themselves in this area. I have attached a marked-up pdf in which I've made some suggestions and some minor corrections for typos / language.

I would suggest that the authors consider making minor additions to cover the following two points (also mentioned in the marked-up pdf).

1) I think the review would really benefit from either an expanded version of the table provided, or from an additional table, in which the authors give practical insight into the advantages and limitations of the different models they describe. Here they might want to summarise information that would be very useful to other researchers when selecting suitable models for their studies, e.g. which models might be more amenable to genetic manipulation (CRISPR-Cas9, gene transduction, siRNA etc), which might be useful / scalable for drug screening etc? Also aspects including ease of establishing, cost etc would be really valuable insights for the non-specialist interested in using such models.

2) A number of cell lines and immortalised primary cell cultures are discussed but one significant omission is the use of NIKS. This cell line has been extensively used to study aspects of HPV biology and transformation, particularly with raft cultures (e.g. Isaacson Wechsler. Journal of Virology Volume: 86 Issue 11 2012). I think the review would benefit from a brief mention at least, of the insights that have been gained from experiments using HPV genome transfection of NIKS and from studying NIKS expressing individual HPV oncogenes, in the context of raft cultures in particular.

Author Response

Reviewer 2:

I enjoyed reading this review of in vitro methods for modelling the tumour microenvironment in HPV-associated cancers and I think it will be a very useful introduction for those who have maybe worked largely with 2D culture but are looking for more physiologically relevant systems. Overall, I found it very informative and clearly the authors have made substantial contributions themselves in this area. I have attached a marked-up pdf in which I've made some suggestions and some minor corrections for typos / language.

Thank you the Reviewer for appreciating the review.

I would suggest that the authors consider making minor additions to cover the following two points (also mentioned in the marked-up pdf).

We thanks the referee for her/his observations and we corrected the minor additions in the revised version of the manuscript. Lines 152-153, 188-192, 422-423  in the revised version

1) I think the review would really benefit from either an expanded version of the table provided, or from an additional table, in which the authors give practical insight into the advantages and limitations of the different models they describe. Here they might want to summarise information that would be very useful to other researchers when selecting suitable models for their studies, e.g. which models might be more amenable to genetic manipulation (CRISPR-Cas9, gene transduction, siRNA etc), which might be useful / scalable for drug screening etc? Also aspects including ease of establishing, cost etc would be really valuable insights for the non-specialist interested in using such models.

We thank the referee for this observation and we provide a new table (Table 3) in order to address the referee concerns. Line 401 in the revised version

2) A number of cell lines and immortalised primary cell cultures are discussed but one significant omission is the use of NIKS. This cell line has been extensively used to study aspects of HPV biology and transformation, particularly with raft cultures (e.g. Isaacson Wechsler. Journal of Virology Volume: 86 Issue 11 2012). I think the review would benefit from a brief mention at least, of the insights that have been gained from experiments using HPV genome transfection of NIKS and from studying NIKS expressing individual HPV oncogenes, in the context of raft cultures in particular.

We thank the referee for this observation and we provided information of NIKS by adding this cell line in Table 1 renamed Table 2 (Line 385 in the revised version) as well as by mentioning the usefulness of this cell line in the revised version of the review.

Lines 237-239 in the revised version: “Other researchers reported the use of HPV-16 episome-containing normal immortalized human keratinocyte line (NIKS) that has been extensively cultured to study some aspects of HPV biology and transformation, particularly on raft culture [68].”

Reviewer 3 Report

The review by Gregorio et al have discussed the implications of tumor microenvironment in HPV related cancers by comparing the in vitro models to complex organotypic models. I have some minor concerns that needs to be addressed before consideration

1) The authors mentioned that HPV 16 and HPV 18 as most virulent genotype that cause majority of cancers. The authors are requested to provide any incidence or any correlation with the tumor grade or stage associated with HPV 16 or HPV 18

2) The authors did mention the role of microenvironment with particular emphasis on fibroblasts. However, do other components such as endothelial cells or neuronal cells are also important is not clear. The authors are requested to comment on this aspect and provide a diagrammatic representation of interplay between various cellular components with HPV.

Author Response

Reviewer 3:

The review by Gregorio et al have discussed the implications of tumor microenvironment in HPV related cancers by comparing the in vitro models to complex organotypic models. I have some minor concerns that needs to be addressed before consideration

1) The authors mentioned that HPV 16 and HPV 18 as most virulent genotype that cause majority of cancers. The authors are requested to provide any incidence or any correlation with the tumor grade or stage associated with HPV 16 or HPV 18

We thank the referee for his/her observation. In order to provide major information on the most virulent genotype we added a new table (Table 1, Line 79 in the revised version) in the revised version of the manuscript. In addition, we integrated in the text more information on the incidence/correlation with the tumor grade or stage associated with HPV 16 or HPV 18.

Lines 60-73 in the revised version: “HPV16 is the type most frequently involved in the development of cervical squamous cell carcinomas (50% of squamous tumors are HPV16 positive) and oropharyngeal cancers (~ 25%), whereas both HPV16 and HPV18 are estimated to account for 35% of cervical adenocarcinomas [11]. A subset of cervical and oropharyngeal cancers has been attributed to the infection with other high-risk HPVs. Recurrent respiratory papillomatosis, a very rare disease, which is difficult to treat and has high recurrence rates, is highly associated with low-risk HPV6/11 (Table 1). HPVs are also responsible for a significant proportion of precancerous lesions of the vulva (vulvar intraepithelial neoplasia grades 2 and 3 -VIN2/3-), vagina (vaginal intraepithelial neoplasia grades 2 and 3 -VaIN2/3-), anus (anal intraepithelial neoplasia grades 2 and 3 -AIN2/3-), penis (penile high-grade squamous intraepithelial lesions), head and neck, as well as genital warts. All Cervical Intraepithelial Neoplasia (CIN) is HPV-related, with HPV6/11/16/18 accounting for 23-25% of CIN1, 38.4-39% of CIN2, and 58% of CIN3 [12]. For VIN2/3, VaIN2/3 and AIN2/3, more than 80% of total number of annual cases, based on the age-specific incidence data, were estimated to be attributable to all HPV types [13].”  

2) The authors did mention the role of microenvironment with particular emphasis on fibroblasts. However, do other components such as endothelial cells or neuronal cells are also important is not clear. The authors are requested to comment on this aspect and provide a diagrammatic representation of interplay between various cellular components with HPV.

We thank the referee for his/her observation. The authors provide a new figure (Figure 3, Line 79 in the revised version) with diagrammatic representation of all cellular components (tumoral and non-tumoral) including neuronal and endothelial cells and their interplay with HPV. In addition, the authors extensively commented this aspect in the revised version of the review.

Lines 351-374 in the revised version: “It's also noted that the carcinogenesis of cervical carcinoma implies an intricate interplay of neoplastic, HPV-infected epithelial cells and stromal tissue including non-tumoral cell types [105]. The HPV-positive epithelium and stromal cells (CAFs, endothelial cells, immune cells and neuronal cells) communicate with HPV-infected epithelium through the exchange of growth factors (TGF-β, VEGF, HB-EGF, EGF), cytokines (IL, CXCLs and CCLs), neurotransmitters, ECM molecules (MMPs) and other molecules (M-CSF, G-CSF), leading stromal remodeling, cancer cells proliferation and angiogenesis processes [27]. A thorough mapping of the non-tumoral cell types that populate the TME is critical to understand their unique roles in tumor biology. Interestingly, tumor innervation is associated with worse clinical outcomes in several solid cancers [106, 107], emphasizing nerves as microenvironmental factors that may contribute to tumor progression. Scientific reports suggest that carcinogenesis alters cervical innervation, demonstrating the role of the HPV-positive cervical cancer cell lines in effectively stimulate neurite outgrowth [108]. Furthermore, an essential component of the tumor-associated stroma is the vasculature, composed of blood and lymph vessels [105]. The induction of angiogenesis is an early event in cervical carcinogenesis [109]. In details, in low-grade lesions, there is an increase in the number of capillaries in the cervical stroma underlying the dysplastic epithelium. In high-grade lesions there is an additional increase in the number of vessels that appear to be organized into a dense micro-vascular array in close relation to the overlying neoplastic epithelium. Furthermore, in some high-grade lesions, stromal vascular papillae are formed that reach towards the surface of the epithelium [110]. Over the past years, the endothelial cells have been used in vitro as feeder for keratinocytes to support epithelial cell differentiation [32]. Other studies reported the HPV-dependent angiogenic response in terms of proliferation and migration of the endothelial cells when cultured with conditioned media from HPV positive cells [111].”

Round 2

Reviewer 1 Report

The paper has been largely improved, but there are still some remaining issues that need to be addressed

The additional information on HPV types and the different diseases as described from lines 54-67 needs some restructuring and may be shortened. E.g. Combine the precursor lesions with the respective cancers and briefly indicate that HPV16 and 18 are the major contributors to cervical cancer, with 30% being caused by other hr-HPV types. In case of the other HPV-induced cancers HPV16 is most commonly involved. Of note, I would say that most CIN lesions are HPV-related, rather than all; HPV-negative CIN have also been reported.

Table 1 can be omitted or also the other HPV-induced cancers need to be included to make it complete

Line 71: adapt to: …..asymptomatic, persistent HPV infections have an increased chance to induce epithelial cell abnormalities ….

Line 74: please correct: a monolayer of squamocolumnar cells is not known in the anus and in the cervix these are referred to as squamocolumnar junction cells

Line 90-93: The sentences are confusing as they may be interpreted as CIN3 lesions directly progressing to metastatic disease. I assume it is meant to describe progression from premalignant disease (CIN2/3) to invasive cancer. This does not necessarily mean metastasis.

Line 93: what is meant with the rest?

Although the newly added lines 73-98 are well appreciated, they may be shortened. Please be aware that although HPV integration is often seen in cancers, it is not evident in all cancers. In some cancers episomal DNA is found.

Line 171: correct episomial.

Lines 172-175 I would like to suggest to reorder a bit. From line 168 onwards productive infection is discussed, then maintenance and then again productive

Line 206: as mentioned before I would not refer to cervical cancer cells as in vitro immortalized cells. In vitro immortalized keratinocytes cell lines are for example primary HFKs transfected or transduced with HPV, that upon continuous culturing become immortalized. See for example: ref 72, 73 and PMC1886282, PMC3911618, PMC1876907, PMID:8808699. Some of the latter also describe 3D raft cultures.

Same for line 212: in vitro immortalized from a cancer cells? Cancer cells are generally immortal by themselves. Ref 52 is on a condyloma not a carcinoma, ref 67 is on primary keratinocytes

Table 3 and Figure 3 are very informative!
